# The CH_3_D Absorption Spectrum Near 1.58 μm: Extended Line Lists and Rovibrational Assignments

**DOI:** 10.3390/molecules29225276

**Published:** 2024-11-08

**Authors:** Ons Ben Fathallah, Anastasiya Lembei, Michael Rey, Didier Mondelain, Alain Campargue

**Affiliations:** 1CNRS, LIPhy, University Grenoble Alpes, 38000 Grenoble, France; ons.ben-fathallah@univ-grenoble-alpes.fr (O.B.F.); belansib@gmail.com (A.L.); didier.mondelain@univ-grenoble-alpes.fr (D.M.); 2GSMA, UMR CNRS 7331, University of Reims Champagne Ardenne, Moulin de la Housse B.P. 1039, F-51687 CEDEX Reims, France; michael.rey@univ-reims.fr

**Keywords:** methane, CH_4_, CH_3_D, Titan, absorption spectroscopy, TheoReTS, HITRAN, rovibrational assignments

## Abstract

Monodeuterated methane (CH_3_D) contributes greatly to absorption in the 1.58 μm methane transparency window. The spectrum is dominated by the 3ν_2_ band near 6430 cm^−1^, which is observed in natural methane and used for a number of planetary applications, such as the determination of the D/H ratio. In this work, we analyze the CH_3_D spectrum recorded by high-sensitivity differential absorption spectroscopy in the 6099–6530 cm^−1^ region, both at room temperature and at 81 K. Following a first contribution to this topic by Lu et al., the room-temperature line list is elaborated (11,189 lines) and combined with the previous 81 K list (8962 lines) in order to derive about 4800 empirical lower-state energy values from the ratio of the line intensities measured at 81 K and 294 K (2*T*-method). Relying on the position and intensity agreements with the TheoReTS variational line list, about 2890 transitions are rovibrationally assigned to twenty bands, with fifteen of them being newly reported. Variational positions deviate from measurements by up to 2 cm^−1^, and the band intensities are found to be in good agreement with measurements. All the reported assignments are confirmed by Ground-State Combination Difference (GSCD) relations; i.e., all the upper-state energies (about 1370 in total) have coinciding determinations through several transitions (up to 8). The energy values, determined with a typical uncertainty of 10^−3^ cm^−1^, are compared to their empirical and variational counterparts. The intensity sum of the transitions assigned between 6190 and 6530 cm^−1^ represents 76.9 and 90.0% of the total experimental intensities at 294 K and 81 K, respectively.

## 1. Introduction

The 1.58 μm transparency window of methane, ranging between about 6200 and 6500 cm^−1^, is a very weak absorption region of particular importance in the study of the giant planets [1,2] and Titan [3,4,5]. This spectral interval has been used for probing very deep into the atmospheres, down to the troposphere and even to the surface. The quality of the methane line parameters used for the simulations of planetary spectra has a direct impact on the information retrieved from astronomical observations. The weak methane absorption in the 1.58 μm window allows the inference of the D/H ratio in methane of the studied atmospheres from the detection of CH_3_D absorption lines of the 3ν_2_ stretching band centered near 6428.4 cm^−1^ [3]. In spite of the low value of the CH_3_D/CH_4_ relative isotopic abundance (about 5 × 10^−4^ in “natural” gas), CH_3_D has a large relative contribution in the region [6] (up to 70% near 6300 cm^−1^ at 80 K—see Figure 1 of Ref. [7]). This is due to the large difference in the CD and CH stretching frequencies, which makes some CH_3_D bands (e.g., 3ν_2_) shift apart in the CH_4_ strong-absorption regions.

Since the 1980s, planetary applications have motivated a number of studies of CH_3_D absorption in the 3ν_2_ region. Lutz et al. [10] and Boussin et al. [11,12] determined the positions, line intensities, and pressure broadening parameters of about 250 transitions from high-resolution spectra recorded by Fourier transform spectroscopy (FTS) using a mostly pure CH_3_D sample at pressure values up to 250 Torr and temperatures of 150 and 295 K. More recently, a global rovibrational analysis of the infrared FTS spectrum of CH_3_D at 80 K in a collisional cooling cell was reported by Ulenikov et al. [13]. In our group, we have applied the cavity ring-down spectroscopy (CRDS) technique to characterize the 1.58 μm transparency window of methane in a natural isotopic concentration both at room temperature and liquid nitrogen temperature (~81 K) [6,14,15]. The sensitivity of the CRDS technique allows for the detection of CH_3_D lines in spite of its very small relative abundance. In order to discriminate the CH_3_D lines from the CH_4_ lines, the CRDS methane spectra were compared to FTS spectra of CH_3_D recorded in similar temperature conditions. Apart from the 3ν_2_ transitions, many additional CH_3_D lines were identified and are tagged in the WKLMC (Wang, Kassi, Leshchishina, Mondelain, Campargue) [9] empirical lists for methane at 80 K and 296 K (see the overview between 5800 and 7000 cm^−1^ presented in Figure 1). The WKLMC lists covering the large 5852–7919 cm^−1^ range were derived from methane spectra recorded by differential absorption spectroscopy (DAS) in the strong absorption regions [16,17] and by CRDS in the 1.58 µm [15] and 1.28 µm [18,19] transparency windows. Note that since its 2012 edition, the HITRAN database has adopted the WKLMC list at 296 K for ^12^CH_4_ above 6100 cm^−1^ [20].

Due to frequent overlapping with CH_4_ lines, the use of spectra of methane in natural isotopic abundance to derive CH_3_D line parameters is not ideal. This is the reason why we performed a series of DAS recordings of spectra of pure CH_3_D at 81 K and 294 K in the 6099–6530 cm^−1^ region. A first report by Lu et al. [7] was dedicated to the analysis of these spectra. A list of about 9000 lines was constructed from the 81 K spectrum. The temperature dependence of the line intensities being necessary for planetary applications, in the absence of rovibrational assignments, we derived empirical values of the lower-state energy (*E_emp_*) from the ratio of the intensities at 81 K and 294 K. As Lu et al. reported line intensities at 294 K only in the 6204–6394 cm^−1^ interval, the 2*T*-method could be applied only in this limited region, and 2723 *E_emp_* values were derived for transitions observed both at 81 and 294 K. For the HITRAN2020 list in the region [8], the lines identified as CH_3_D were removed from the WKLMC list, and the room-temperature CH_3_D list obtained by Lu et al. [7] was adopted in the 6204–6394 cm^−1^ interval (see the lower panel of Figure 1). (Note that the removal of the contribution of a minor isotopologue from an empirical line list obtained with a natural sample is not easy and may be problematic in the case of a congested spectrum, with some empirical lines being blended due to the superposition of lines belonging to different isotopologues [9].)

The aim of the present contribution is to extend the analysis of the DAS spectra of Lu et al. by two means. First, the room-temperature line list of Lu et al. limited to the 6204–6394 cm^−1^ interval will be extended to coincide with the range of the 81 K list (6099–6530 cm^−1^). Line parameters will be retrieved in the 6099–6204 cm^−1^ strong-absorbing interval and in the 6394–6530 cm^−1^ region, including the *Q*- and *R*-branches of the 3ν_2_ band (see Figure 2). As a result, the 2*T*-method will allow the derivation of *E_emp_* values for the whole range. Second, the rovibrational assignments of both the 81 K and 294 K spectra will be considered on the basis of the TheoReTS ab initio line list. The TheoReTS variational lists of the various isotopologues of methane were calculated from extensive first-principle calculations using accurate ab initio potential and dipole moment surfaces [21,22,23]. The current quality achieved by this type of calculations makes it possible to reliably assign the near-infrared absorption spectra of medium-size molecules such as C_2_H_4_ [24] or NH_3_ [25]. In spite of a high spectral congestion and the absence of regular sequences of absorption lines, a reasonable agreement between the measured and calculated positions and intensities, combined with a systematic validation using Lower-State Combination Difference (GSCD) relations, allows for the unambiguous assignments of most of the strong and medium lines. In the present work, where spectra are available at two temperatures, the consistency between the *E_emp_* values derived by the 2*T*-method [14] and the exact lower-state energy obtained from the assignments provide an additional means of checking.

## 2. Experimental Details, Line List Construction, and 2*T*-Method

An overview of the CH_3_D spectra at 294 K and 81 K is displayed in Figure 2. The experimental setup for differential laser absorption spectroscopy and the spectrum acquisition procedure have been described in detail in Refs. [7,14] and will not be repeated here. Briefly, the spectral region extending between 6099 and 6991 cm^−1^ was covered with the help of a series of thirty-seven DFB fibered diode lasers, whose emitted light was distributed between the absorption cell, a home-made etalon, and a wavelength meter, with these last two being used for the frequency calibration of the spectra. The spectrum over an entire DFB range of about 35 cm^−1^ was typically recorded within 12 min by the slow tuning of its emitted frequency using a ramp between −10 °C and +60 °C for the diode temperature. One of the two weak beams reflected from the input cell window was used as a reference to correct the variations in the laser power. The 1.42 m long cryogenic cell was used in a round-trip configuration (absorption pathlength *L* = 284 cm) and filled with 10 Torr of high-purity CH_3_D (min 98 atom % D). As detailed in Lu et al., the frequency scale of each spectrum corresponding to a given DFB laser was linearized independently with the help of the etalon fringes. The absolute frequency calibration was obtained by statistical position matching with accurate CH_3_D line positions retrieved from an FTS spectrum of CH_3_D at room temperature, with the latter being itself calibrated against line positions of H_2_O (present as an impurity in the CH_3_D cell). As a result, the uncertainty of the absolute values of the centers of the well-isolated lines is estimated to be on the order of 1 × 10^−3^ cm^−1^.

The noise-equivalent absorption (NEA) of the spectra, on the order of *α_min_* ≈ 5 × 10^−8^ cm^−1^, corresponds to an intensity threshold on the order of 5 × 10^−26^ cm/molecule at room temperature. Note that at 81 K, the detectivity threshold is at the 5 × 10^−27^ cm/molecule level due to the two-times-smaller Doppler width and four-times-larger molecular density at 81 K.

In Figure 3, we compare the 294 K spectrum to the 81 K spectrum in a small spectral section where the variation in the intensities by cooling is particularly pronounced.

As explained above, the first step was to construct the room-temperature line list in the 6099–6204 cm^−1^ and 6394–6530 cm^−1^ intervals in order to coincide with the range of the 81 K line list of Ref. [7] (6099–6530 cm^−1^). The two studied intervals correspond to the successive and partly overlapping ranges of four and six DFB laser diodes, respectively. (Note that two small spectral gaps, 6132.3–6133.2 cm^−1^ and 6201.6–6204.0 cm^−1^, were not accessible with the available laser diodes).

A home-made interactive multi-line fitting program was used to reproduce the spectrum and derive the line parameters. A Voigt function was adopted for the line shape; the width of the Gaussian component was fixed at the theoretical value of the Doppler broadening (≈10^−2^ cm^−1^ HWHM at 296 K). The line intensity of a rovibrational transition centered at *ν*_0_, *S* (cm/molecule), was obtained from the integrated line absorbance and the molecular concentration calculated from the measured pressure value. The combined line lists obtained in the 6099–6204 cm^−1^ and 6394–6530 cm^−1^ intervals include about 2200 and 3550 lines, respectively. The first interval corresponds to the high-energy range of the very strong tetradecad bands of ^12^CH_4_. The comparison to the WKLMC methane line list [9] provided by the HITRAN2020 database [8] allowed us to identify about sixty ^12^CH_4_ lines, which were removed from the list. From the line intensities, a relative concentration of about 1.5 × 10^−3^ was estimated for ^12^CH_4_ compared to CH_3_D. The resulting cleaned line lists were merged with the 294 K list obtained by Lu et al. for the 6204–6394 cm^−1^ region, leading to a global list of 11,189 lines for the entire 6099–6530 cm^−1^ region at room temperature (the upper panel of Figure 4). In the experiment, which will now be elucidated, the 2*T*-method was applied to the present room-temperature line list and to the 81 K list elaborated by Lu et al. [7]. It should be noted that about 70 ^12^CH_4_ lines were identified in the 81 K list below 6160 cm^−1^. After the removal of these ^12^CH_4_ lines, the resulting 81 K list included 8962 CH_3_D lines (the lower panel of Figure 4).

The temperature dependence of the line strengths is ruled by the Boltzmann factor:(1)$$S_{T}(T)\propto \frac{\exp(-E/k_{B}T)}{Z(T)}$$
where *Z*(*T*) is the partition function and *E* is the lower-state energy level. An empirical value of the lower-state energy, *E_emp_*, of a given transition can be determined from the ratio of the line intensities measured at two temperatures [14]:(2)$$E_{emp}\left[\frac{1}{kT_{1}}-\frac{1}{kT_{0}}\right]=\ln\left(\frac{Z(T_{0})S_{v_{0}}(T_{0})}{Z(T_{1})S_{v_{0}}(T_{1})}\right)$$
where *T*_0_ = 294 K and *T*_1_ = 81 K. The values of the partition function at 81 K and 294 K, as given in the HITRAN database, lead to a *Z*(294 K))/*Z*(81 K) ratio of 6.876 (the $T^{3/2}$ approximation of the temperature dependence of *Z*(*T*) leads to a ratio of 6.915).

The lines of the 81 K and 294 K datasets were associated automatically using the coincidence of their positions: two lines were considered as corresponding to the same transition when the difference in their centers differed by less than 0.003 cm^−1^. According to this criterion, 4800 pairs of coinciding lines were found, allowing for the same number of *E_emp_* determinations (this number includes the 2723 *E_emp_* values already determined in Lu et al. for the 6204–6394 cm^−1^ central interval). The associated lines are highlighted with colored symbols in the overview plots presented in Figure 4. The *E_emp_* values span up to about 730 cm^−1^. As an example, the *E_emp_* values are indicated on the spectra displayed in Figure 5, where lines showing a large variety of temperature dependences are located. The complete list at 294 K provided in the Appendix A includes, for each line, the position and intensity at 294 K, together with those of the coincident line at 80 K (when available) and the corresponding *E_emp_* value. A similar Appendix A list is provided for the 81 K temperature.

Overall, the associated transitions represent 85.8% and 95.0% of the total absorption at 294 K and 81 K, respectively. Thus, although the number of derived *E_emp_* values represents roughly half the number of lines measured at the two temperatures, the temperature dependence of most of the absorption of the region is well characterized, with the lines without *E_emp_* value being weak or very weak (see Figure 5). It should be noted that for the CH_4_ species, the values of the lower-state energy are well separated and the accuracy of the determined *E_emp_* values allows for the unambiguous determination of the *J* quantum number (e.g., Ref. [14]). This is not the case in CH_3_D, for which the values of the rotational energies are too dense to unambiguously derive the empirical values of the *J* and *K* quantum numbers. In the following part, we will present a comparison of the *E_emp_* values with their exact values obtained from the rovibrational assignments provided by theory.

## 3. Spectra Analysis

### 3.1. Rovibrational Assignments

As illustrated by the rovibrational assignments of 108 sub-bands of ^12^CH_4_ in the icosad region (6280–7800 cm^−1^) [26], the quality of the TheoReTS methane lists [27,28] allows for a reliable assignment of the methane spectrum in the near-infrared region. The TheoReTS lists were obtained from extensive first-principle calculations using an accurate ab initio potential. The CH_3_D list [22], available at http://theorets.tsu.ru accessed on 10 April 2023, is provided at 296 K in the 0–6500 cm^−1^ range with an intensity cut-off of 10^−25^ cm/molecule. For the purpose of the present work, two line lists at 81 K and 294 K were calculated using the same PES and DMS as in Ref. [22], and the rovibrational assignments were added to the calculated transitions (only the lower-state energy values are provided in the TheoReTS methane lists available online). In addition, due to the sensitivity of the analyzed recordings, the intensity cut-off was lowered to 10^−26^ and 10^−27^ cm/molecule at 294 K and 81 K, respectively. It is worth mentioning that the labeling of the different levels includes the (v1v2v3v4v5v6 Cv) vibrational assignment corresponding to the quantum numbers of the dominant term in the normal mode expansion of the eigenstate and the vibrational symmetry and the (*J*, *C*, *n*) rotational assignment formed by the *J* rotational quantum number, the rotational symmetry (*C*), and a ranking number, *n*, increasing with the energy, allowing distinction among levels with the same (*J*, *C*) value. Due to the considerable vibrational mixing in the considered energy region, the vibrational labeling may be ambiguous in the sense that distinct upper vibrational levels may have identical vibrational labeling. In this situation, we cannot exclude the possibility that transitions reaching distinct upper vibrational states are gathered under the same band name.

The assignment of the spectrum relies on the position and frequency matchings between theory and the experiment, thoroughly validated by lower-state combination difference (GSCD) relations. The overview comparison between the experimental and variational lists at room temperature is presented in the upper panels of Figure 6. The bird’s-eye view of the entire frequency range shows a good agreement, but on the enlarged scale, in the lower panels, the one-to-one correspondence between the experimental and calculated transitions is less obvious, in particular for the weak lines. The observed position differences between the TheoReTS and experimental line centers (up to 2 cm^−1^) make it necessary to systematically validate the rovibrational assignments with the help of GSCD relations, i.e., to check that the TheoReTS position shift of all the transitions reaching a given upper level is identical. In practice, the assignment procedure consisted of different steps:

(i) For the accurate application of the GSCD relations, very accurate values of the lower-state energy (*E_low_*) have to be used. For this purpose, the variational *E_low_* values of the initial TheoReTS list were replaced by their HITRAN values (differences up to 0.03 cm^−1^ were noted for *E_low_* values on the order 600 cm^−1^).

(ii) The transition lists of the different bands of the TheoReTS list contributing to the spectrum were sorted. Overall, we considered a total of 20 bands, listed in Table 1 and Table 2.

(iii) Starting from the dominant band, each TheoReTS band was considered successively in decreasing order of their intensity. For a given experimental line, based on the position and intensity agreement, a counterpart was tentatively identified among the nearby variational transitions of the considered band. We will call *δ*_0_ = (ν_exp_. − ν_calc_.) the corresponding position shift. Using a dedicated home-made program developed under LabVIEW, all the TheoReTS transitions reaching the upper state of the candidate transition were identified in the calculated line list and searched in the experimental list with identical position shift (*δ*_0_). The tentative assignment was considered as validated when all the predicted transitions (with sufficient intensity) were observed in the spectrum and showed an identical *δ*_0_ value, with a reasonable intensity agreement (a conservative value of 5 was tolerated for the intensity ratio of the experimental and variational intensities, the intensity agreement being generally much better—see below). More precisely, taking into account the experimental position uncertainty, the GSCD relations were considered as being fulfilled when the difference between the experimental value of the wavenumbers of two transitions reaching a given upper level coincided with the difference in the corresponding lower-state energies within 4 × 10^−3^ cm^−1^ (this value corresponds thus to the maximum difference on *δ*_0_ for the set of transitions sharing the same upper state). The different transitions reaching a given upper level allow for various (up to eight) coincident determinations of the upper-state energy.

As an example, the assigned transitions of the 3ν_2_, ν_1_ + 3ν_6_, and ν_2_ + ν_4_ + ν_6_ bands are highlighted on the experimental and variational stick spectra displayed in Figure 7. A variational counterpart has been found for most of the observed lines with significant intensity, and vice versa.

An overview of all the assignments is presented in Figure 8 for the whole region. The assigned transitions belong to twenty bands, fifteen of them being newly reported (see the literature review below). For each band listed in Table 1 and Table 2, the total number of transitions included in the TheoReTS list is given, together with the number of assigned lines, the maximum values of the *J*, *K* rotational quantum numbers, and a comparison of the sum of their intensities. Overall, the intensity agreement is good both at 80 K and 294 K: experimental and variational band intensities coincide within 10% for twelve of the twenty bands, while the total sums of the intensities of the assigned transitions are in perfect agreement. It should be noted that about 2720 transitions are assigned at both temperatures, compared to a total number of measured lines of 8962 and 11,189 at 81 K and 294 K, respectively. Nevertheless, the large number of unassigned transitions corresponds to no more than 10% and 23% of the total experimental intensities at 81 K and 294 K, respectively. The unassigned lines most likely include a significant number of ^13^CH_3_D lines as, assuming there is a ^13^CH_3_D/^12^CH_3_D relative abundance of about 1% corresponding to the usual ^13^C/^12^C value, the strongest ^13^CH_3_D lines in the region are expected to have an intensity on the order of a few 10^−25^ cm/molecule, largely above the experimental detectivity threshold of a few 10^−26^ cm/molecule.

The rovibrational (RV) assignments have been added to the two experimental lists provided in Appendix A. A sample of the 81 K list is presented in Figure 9. We have gathered, in the two lists, the spectroscopic information provided by the 2*T*-method and by the RV assignments based on TheoReTS. In a few situations, a transition could be assigned at one temperature but not at the other because of the change in the spectrum appearance, which prevented the observation of one transition involved in GSCD relations (for instance, high rotational transitions are missing in the 81 K dataset). In those situations, the RV assignment was transferred from one temperature to the other. In addition, we extrapolated the intensity values from 81 K to 294 K using the *E_low_* values and checked the consistency of the extrapolated value by a direct superposition to the spectrum at 294 K. The same test was performed by extrapolating the 294 K line list at 81 K. In the few cases of strong disagreement between the extrapolated line intensity and observation, the RV assignment was deleted. In a number of cases, the temperature dependence of the intensity was validated by all but one transition involved in GSCD relations. In those situations, the RV assignment of the problematic component was kept but tagged with “d” (doubtful).

In Table 3, the number of lines with RV assignments and known *E_emp_* values is given for 81 K and 294 K. (The statistics are limited to the 6190–6530 cm^−1^ region, because we did not attempt to assign the strong lines located below 6200 cm^−1^, as they belong to bands centered below 6100 cm^−1^, for which only the high-energy part of the rotational structure is observed, hampering the application of GSCD relations.) For about 2300 lines, both the lower-state energy (*E_low_*) and *E_emp_* are known. As an example, in Figure 10, the *E_low_* and *E_emp_* values are indicated near each line. Instead of a direct comparison of the *E_low_* and *E_emp_* values, it is more relevant to check the agreement of the corresponding Boltzmann factors (Equation (2)). For the 294 K dataset, 85% of the transitions show an agreement within 20% of the Boltzmann factors calculated with *E_low_* and *E_emp_* values.

### 3.2. Energy Levels

The 81 K and 294 K datasets allow for the determination of 11,868 and 1289 upper-state energies, respectively, by adding the lower-state energy to the measured line positions. Due to the GSCD criterion, up to eight determinations are available for each upper level. The number of determinations, average energy value, and corresponding RMS are provided separately in Appendix A. Over the 11,868 upper energies derived from the 81 K spectrum, 757 correspond to an RMS value less than 1 × 10^−3^ cm^−1^. In the case of the 294 K dataset, over a total of 1289 levels, 672 and 969 levels have their energy determined with a precision better than 1 × 10^−3^ cm^−1^ and 2 × 10^−3^ cm^−1^, respectively. These values reflect the precision of the line center determinations, which is about twice as good at 81 K as at 294 K (due to the two-times-smaller Doppler line width). The mean energy difference for the 1111 levels in common at 81 K and 294 K is 8.21 × 10^−5^ cm^−1^ with an RMS of 0.00107 cm^−1^ (excluding two energy levels for which we could not decide between distinct energy values derived at 81 K and 294 K; these two values are identified in the Appendix A).

## 4. Discussion

### 4.1. Comparison Between Experiment and Theory

The differences between the variational and experimental energy values are plotted in Figure 11 versus the average energy of the upper states, for a total of eleven bands. For the displayed bands, a similar behavior is noted: overall, the variational energies are underestimated, with differences increasing from a small value (less than 0.2 cm^−1^) up to about 1 cm^−1^ for rotational energies on the order of 600 cm^−1^. Thus, the vibrational energies are satisfactorily predicted for the considered bands. In order to improve the accuracy of the TheoReTs line positions in the region, an empirical law based on the average deviations of the rotational energies could be used for the empirical correction of the ThoeReTS rotational energies of all the bands in the region. Regarding the rotational dependence, it is very similar to that observed for C_2_H_4_ bands in the same spectral region [24]. The general increase in the amplitude of the deviations with the rotational energy was tentatively interpreted as being due to an insufficient convergence of the variational calculations for high *J* values. We should note that the dispersion of the deviations around the general tendency largely exceeds the typical 10^−3^ cm^−1^ uncertainty for the determined empirical energy levels. These dispersed values might be due to local resonance interactions between nearby rovibrational levels, which are very difficult to predict accurately.

Interestingly, all but the 3ν_2_ band show a roughly similar rotational dependence for the (Exp.—Var.) position differences. The particular case of the 3ν_2_ band is discussed in detail in the next paragraph, in relation to previous analyses of this particular band available in the literature.

### 4.2. Comparison with Previous Works

As mentioned above, the line positions and line intensities of the 3ν_2_ band near 6430 cm^−1^ were obtained by Lutz et al. [10] and Boussin et al. [11], respectively, from FTS spectra. A systematic underestimation on the order of 0.003 cm^−1^ of the line positions reported by Lutz et al. has been evidenced in Ref. [7]. Regarding line intensities, the FTS values of Ref. [11] were retrieved from a spectrum recorded at 293.5 K with a 10.0 Torr pressure. The comparison of our intensity values at 294 K with the 251 intensities reported by Boussin et al. shows an excellent agreement, with, for instance, an agreement within 1% for the intensity sum.

Using GSCD relations, Lutz et al. [10] were able to rotationally assign 217 transitions of the 3ν_2_ band with (*J_max_*, *K_max_*) = (14, 6) [to be compared to 282 lines with (*J_max_*, *K_max_*) = (14, 12) in the present work]. The rotational structure was found to be affected by perturbations, with an unidentified perturber leading to position shifts, anomalous intensities, and extra lines. A total of 34 transitions belonging to this unidentified band (called “6425 prime” in Ref. [10]) intermixed with the 3ν_2_ band were rotationally assigned using GSCD relations (see Figure 6 of Ref. [7]). On the basis of TheoReTS predictions, the “6425 prime” band is presently assigned to the *ν*_1_ + 3*ν*_6_ (*A_1_*) band, and 63 transitions are assigned on the basis of the 81 K spectra. In Ref. [7], the comparison of our *E_emp_* values with the low-energy values corresponding to the RV assignments of Ref. [10] allowed the testing of the RV assignments. A small number of discrepancies were evidenced. Here, a more complete comparison was possible using our RV assignments based on TheoReTS. Overall, over a total of 251 RV lines, 196 assignments were found to be in perfect coincidence. For 21 lines, the vibrational labeling of the upper state differs, but the upper and lower rotational levels are identical. For the remaining 34 transitions, we propose the correction of the RV assignments of Ref. [10].

Part of the differences in the vibrational labeling is believed to be related to the interaction between the upper states of the 3ν_2_ and *ν*_1_ + 3*ν*_6_ *A*_1_ bands. The position and intensity comparisons with the TheoReTS predictions presented in Figure 12 show that the TheoReTS predicted value of the 3ν_2_ band center is underestimated by about 2 cm^−1^, while the line intensities of part of the transitions of the *ν*_1_ + 3*ν*_6_ *A*_1_ band appear to be smaller than observed by a factor of about two. The underestimation of the vibrational energy of the (030000) state leads to a larger energy separation with the (100003) interacting level, which is consistent with an intensity transfer smaller than observed. We believe that the lack of convergence of the variational calculations is not the main reason for the relatively large difference between the measured and variational energies. The insufficient quality of the potential energy surface (PES) is probably the main explanation. In the present situation of resonance interaction, small inaccuracies in the PES may have an important impact on the calculated energy values.

In Ref. [13], Ulenikov et al. reported 255 rovibrationally assigned lines measured at about 80 K. The assigned transitions belong to five bands: 2ν_4_ *E*, 3ν_2_ *A*_1_, ν_1_ + ν_2_ + ν_6_ *E*, ν_2_ + 2ν_5_ + ν_6_ *E*, and ν_2_ + ν_4_ + ν_6_ *A*_1_. Excluding the 2ν_4_ *E* band located at the low-energy border of the studied region and not considered in our work, our RV assignments of the four remaining bands (239 transitions in total) coincide, except for 19 transitions. For eight of them, only the vibrational labeling differs, leaving only eleven conflictive assignments.

## 5. Conclusions

On the basis of spectra previously recorded by high-sensitivity differential absorption spectroscopy, a room-temperature line list has been elaborated for CH_3_D in the 6099–6530 cm^−1^ region. This list of 11,189 lines includes, in its central part (6204–6394 cm^−1^), about 5000 lines previously measured in Ref. [7]. Based on a comparison with the TheoReTS variational list, 2800 transitions could be rovibrationally assigned to twenty bands. Fifteen bands are newly reported, and the rotational assignments are significantly extended for the others (the maximum value of the *J* quantum number is 15). Ground-State Combination Difference relations were systematically used to validate the assignments. Overall, a total of about 1300 upper-state energies are determined, with a typical error bar of 1 × 10^−3^ cm^−1^. The TheoReTS predictions are found to be in very good agreement with the experiment: vibrational term values deviate from the experiment by less than 0.2 cm^−1^ except for the upper state of the 3ν_2_ band, for which a 2 cm^−1^ underestimation is noted. The obtained RV assignments combined with empirical values of the lower-state energy determined by the *2T*-method will allow us to account for the temperature dependence of most of the absorption lines in the region.

Beyond the specific interest in the 1.58 µm window of methane for planetary applications, the procedure presently used to assign the CH_3_D spectrum by comparison to the TheoReTS line list could be easily applied to other spectral regions. In particular, no data are provided in the HITRAN database for the large 4550–6200 cm^−1^ spectral interval. The quality of the variational predictions should allow us to reliably assign most of the absorption lines using a single spectrum at room temperature.

## Figures and Tables

**Figure 1 molecules-29-05276-f001:**
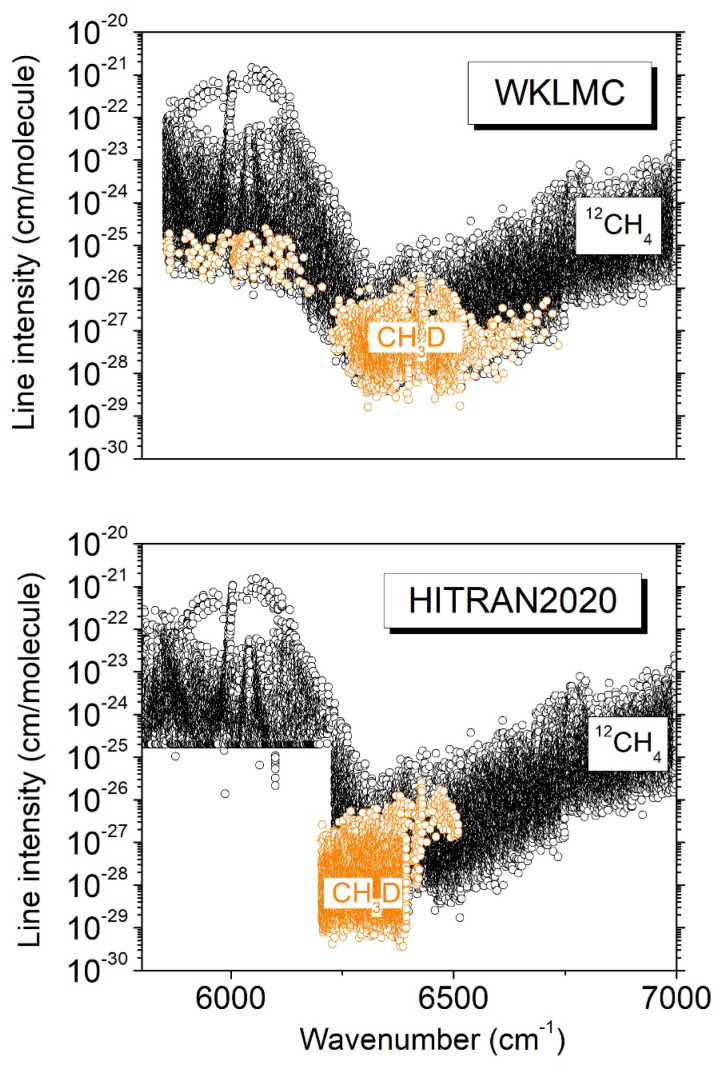
Overview comparison of the room-temperature HITRAN2020 [8] and WKLMC [9] line lists of methane between 5800 and 7000 cm^−1^. The transitions due to CH_3_D in natural isotopic abundance (5 × 10^−4^) are highlighted.

**Figure 2 molecules-29-05276-f002:**
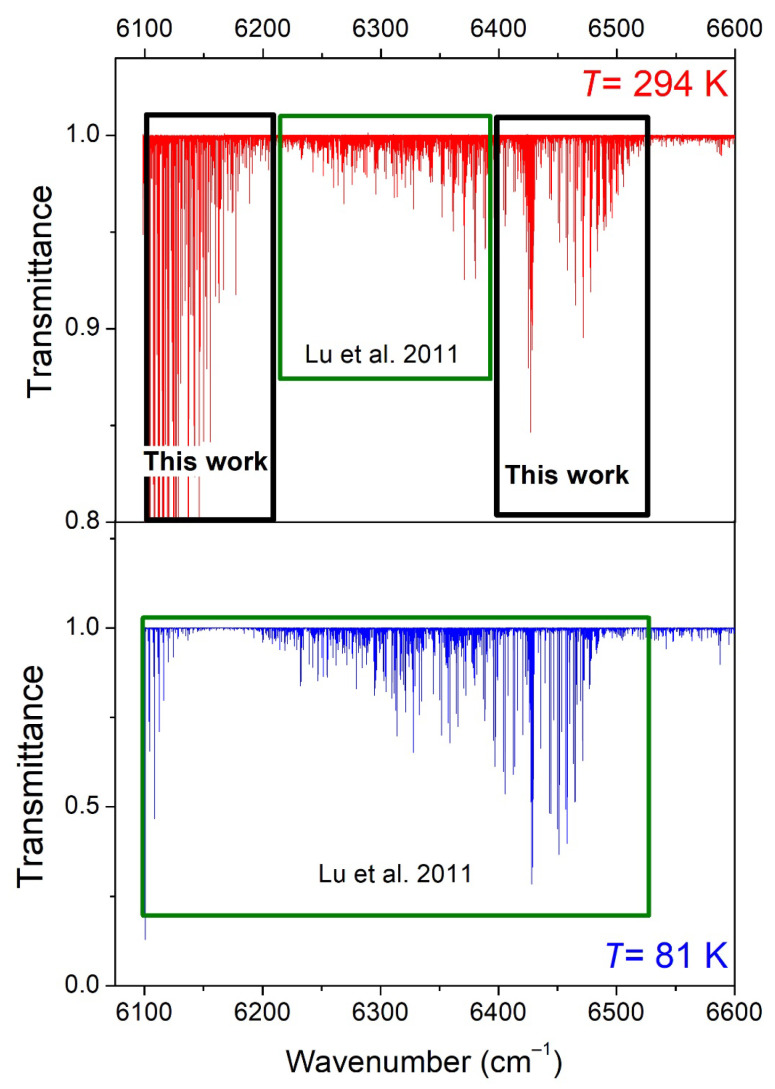
Overview comparison of the absorption spectrum of CH_3_D at 294 K (**upper panel**) and 81 K (**lower panel**) in the 6099–6600 cm^−1^ region. The sample pressures were 10.0 Torr and about 6 Torr, respectively, and the absorption pathlength was 284 cm. In the present work, the room-temperature line list was constructed in the 6099–6204 and 6394–6530 cm^−1^ intervals (black rectangles), while for the region in between, the empirical list was previously elaborated in Lu et al. [7].

**Figure 3 molecules-29-05276-f003:**
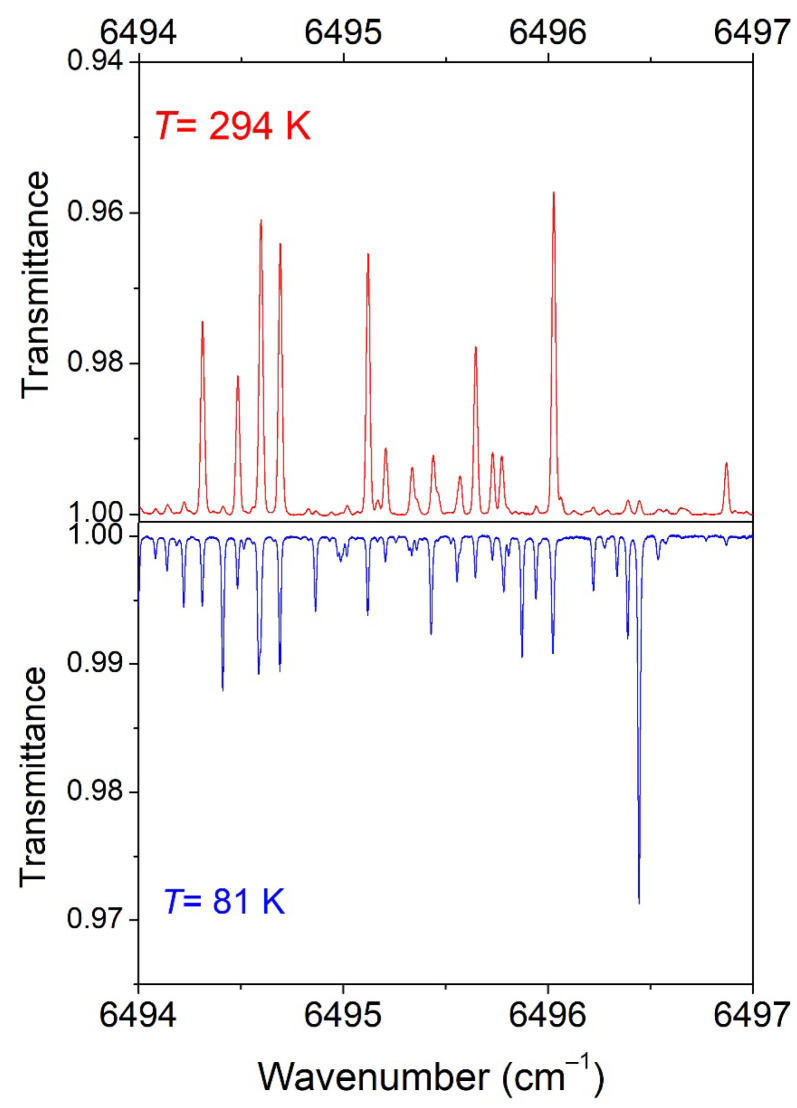
Overview comparison of the CH_3_D spectrum at 294 K (*P* = 10 Torr) and 81 K (*P* = 5.75 Torr) in a spectral region showing a large temperature dependence of the line intensities.

**Figure 4 molecules-29-05276-f004:**
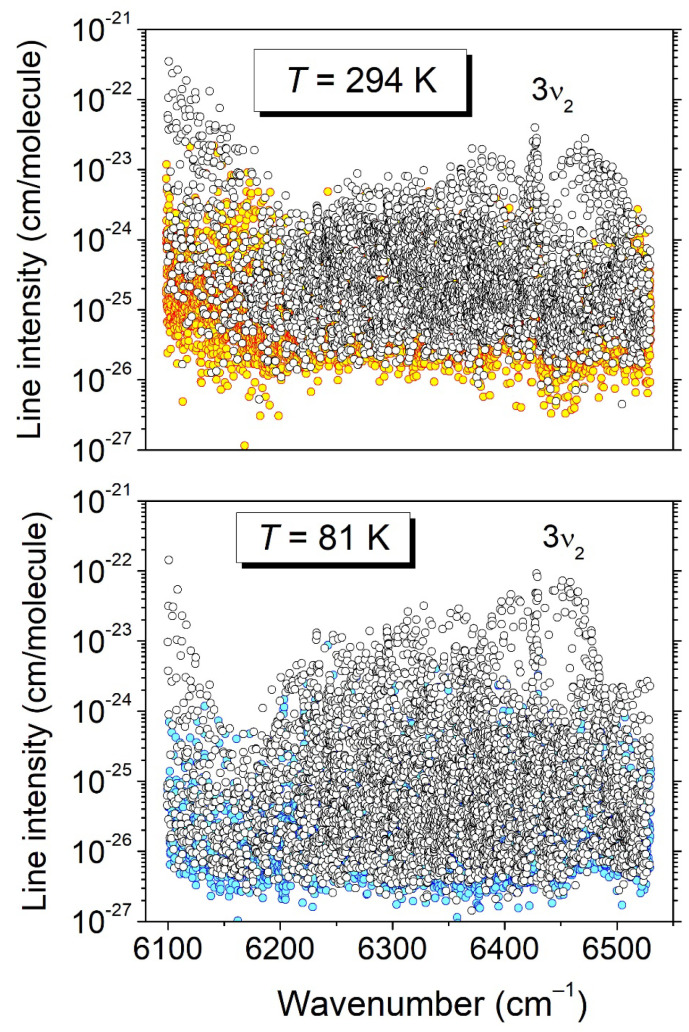
Overview of the CH_3_D line lists at 294 K and 81 K between 6099 and 6530 cm^−1^. In each panel, the colored circles highlight the transitions for which the empirical value of the lower-level energy was derived from the ratio of the line intensities (2*T*-method).

**Figure 5 molecules-29-05276-f005:**
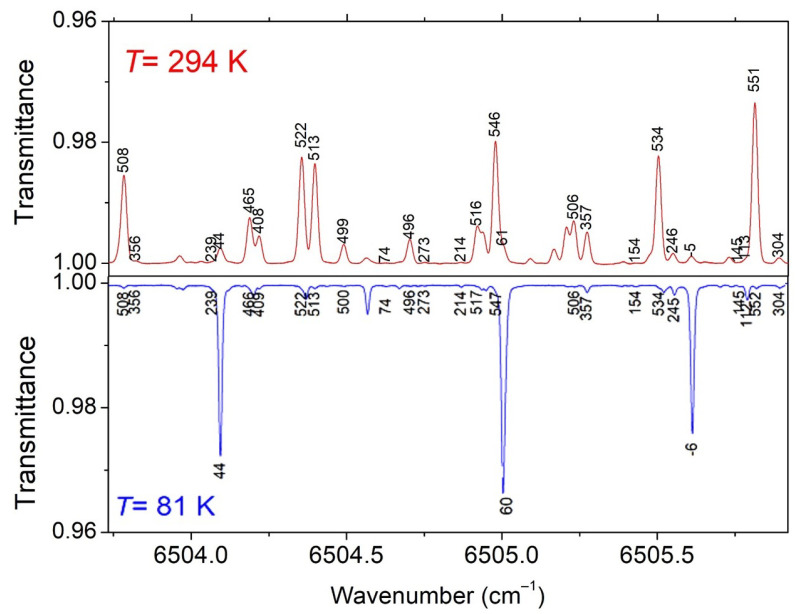
Comparison of the differential absorption spectrum of CH_3_D at 81 K and 294 K near 6505 cm^−1^. The empirical values of the low energy levels (in cm^−1^) derived using the 2*T*-method are indicated.

**Figure 6 molecules-29-05276-f006:**
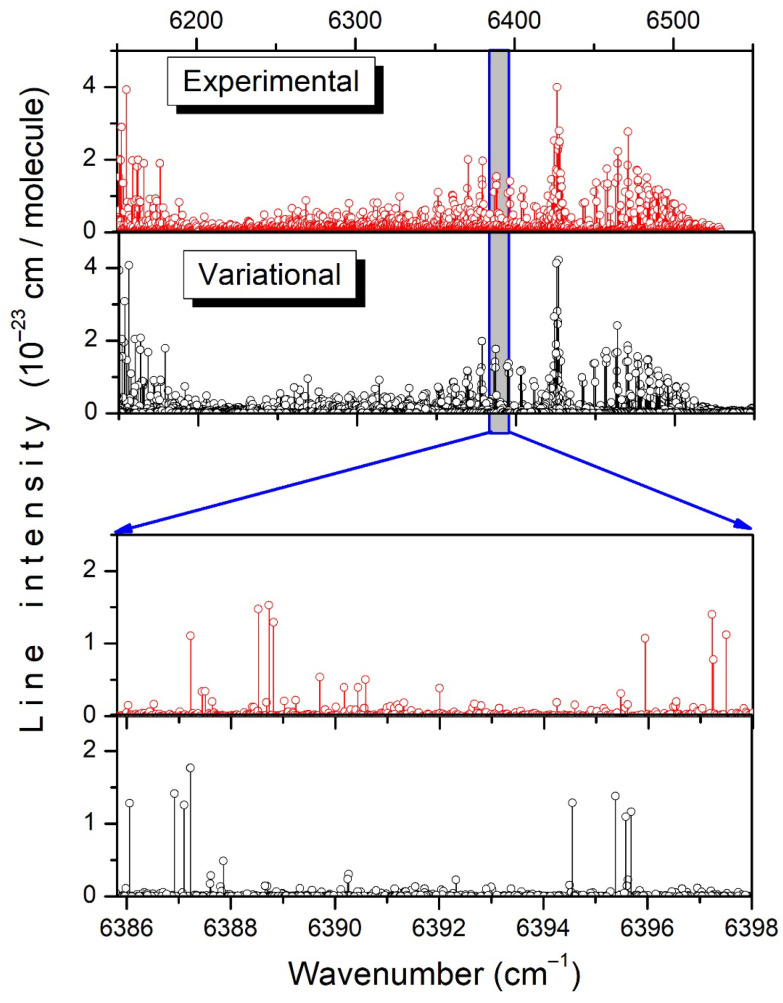
Overview comparison of the experimental and TheoReTS line lists of CH_3_D at room temperature. The lower panels illustrate the level of agreement between the measurements and the calculations.

**Figure 7 molecules-29-05276-f007:**
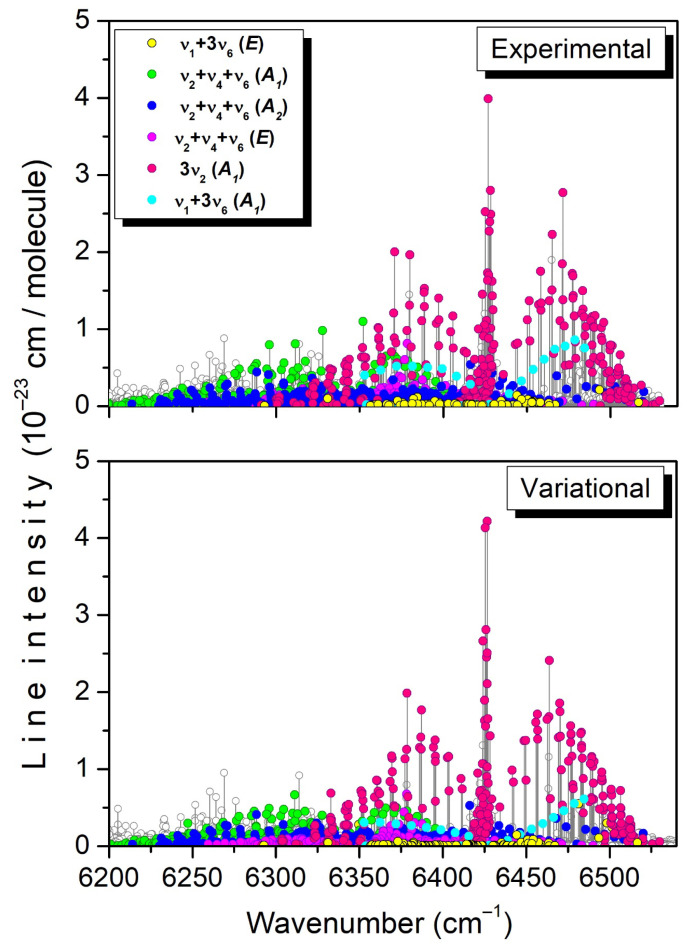
Overview comparison of the experimental and TheoReTS room-temperature lists in the region of the 3ν_2_, ν_1_ + 3ν_6_, and ν_2_ + ν_4_ + ν_6_ bands. The assigned transitions are highlighted with different colors.

**Figure 8 molecules-29-05276-f008:**
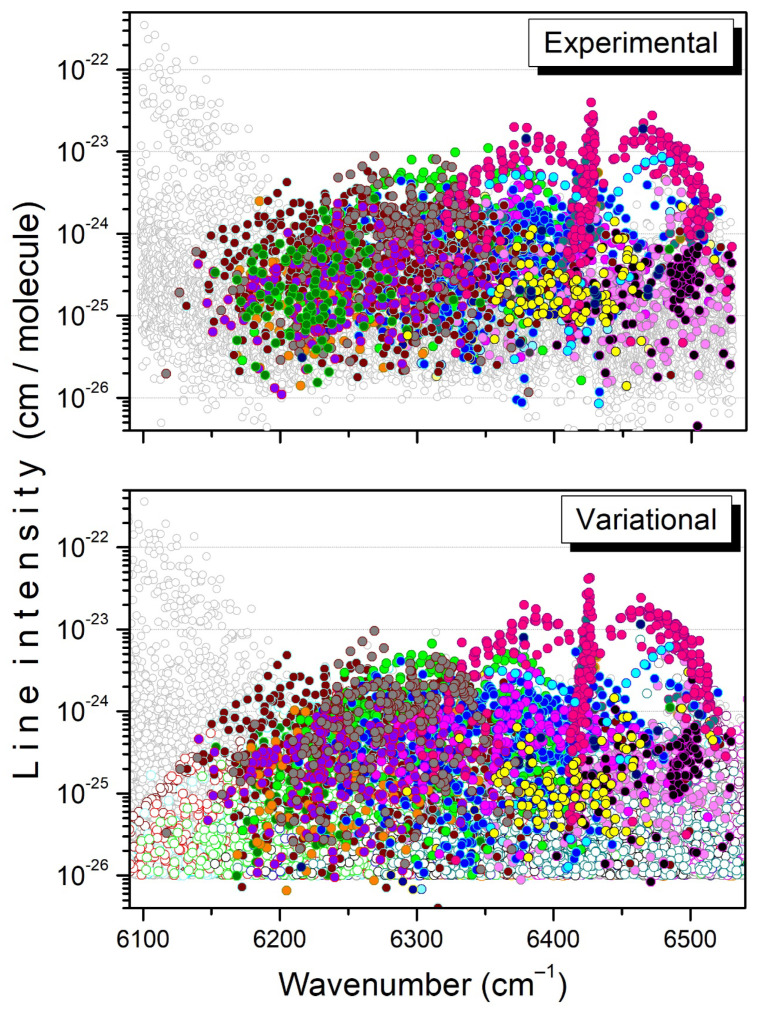
Overview of the experimental and variational line lists at 294 K in the 6090–6530 cm^−1^ interval. Twenty bands predicted by variational calculations are plotted with colored open circles in the lower panel. For each band, the assigned transitions are highlighted with filled circles in both the experimental and variational line lists.

**Figure 9 molecules-29-05276-f009:**
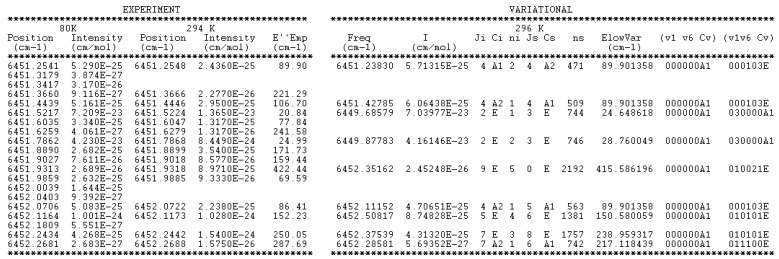
Sample of the 81 K line list of CH_3_D provided in Appendix A. The two first columns correspond to the measured values of the positions and intensities at 81 K. In the case of a position coincidence within 3 × 10^−3^ cm^−1^ with a line measured at 294 K, the empirical value of the lower state provided by the 2*T*-method is given. On the right side of this Table, the variational positions and intensities are given, together with the RV quantum numbers and symmetry of the lower and upper states.

**Figure 10 molecules-29-05276-f010:**
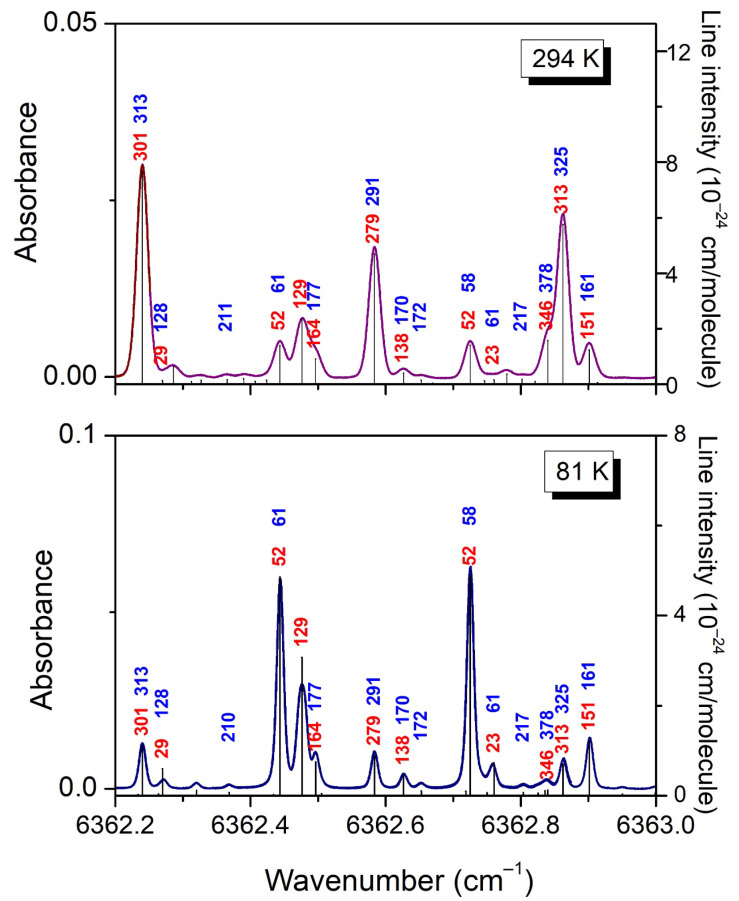
Comparison of the energy of the lower state obtained independently using the 2*T*-method (blue) and obtained from rovibrational assignments (red). The same values are displayed near the lines of the differential absorption spectrum of CH_3_D at 294 K and 81 K (upper and lower panel, respectively).

**Figure 11 molecules-29-05276-f011:**
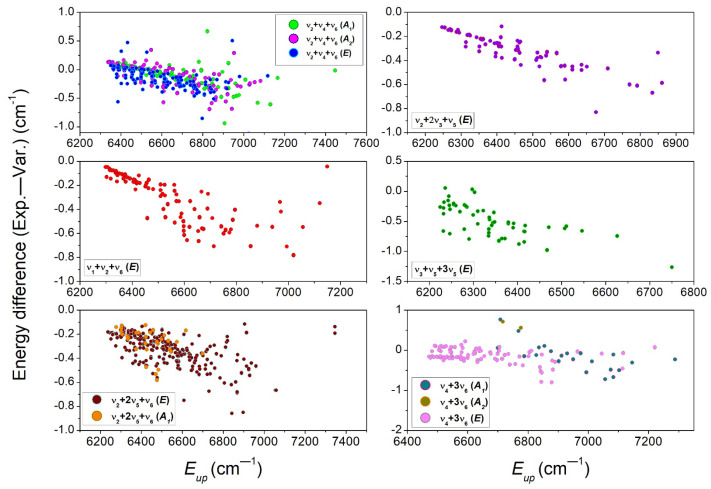
Differences between the empirical and variational values of the upper energy levels versus the upper-state empirical energy for eleven of the twenty ^12^CH_3_D bands assigned between 6099 and 6530 cm^−1^.

**Figure 12 molecules-29-05276-f012:**
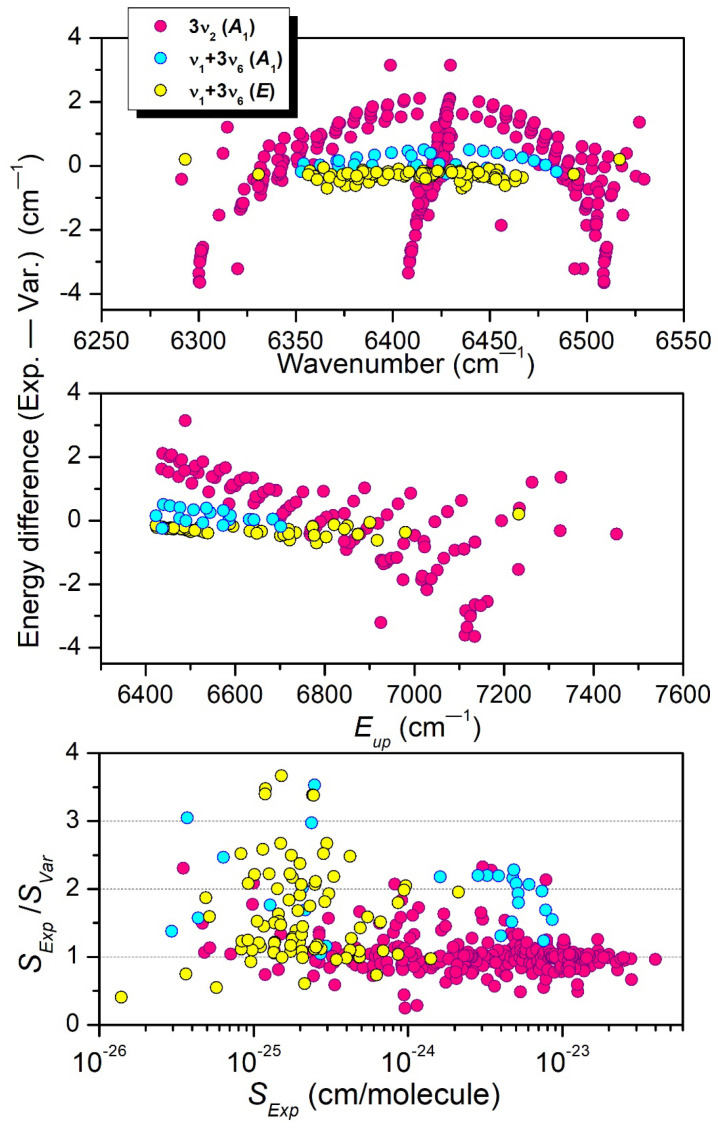
Comparison between experiment and theory for the 3*ν*_2_ and *ν*_1_ + 3*ν*_6_ (*A*_1_) interacting bands. From top to bottom: (Exp.—Calc.) line position differences versus the position value, (Exp.—Calc.) differences of the upper-level energies versus the upper-level energy value, and Exp./Calc. ratio of the line intensities at 294 K versus the measured line intensity.

**Table 1 molecules-29-05276-t001:** Statistics of the transitions assigned to twenty bands in the absorption spectrum of ^12^CH_3_D between 6090 and 6530 cm^−1^ at 80 K and comparison of the measured and calculated intensities.

		Variational			Assigned Lines						
Band	Sym.	Band Center (cm^−1^) ^a^	Nb. ^b^	Int. Sum 296 K ^c^ (cm/Molecule)	Number		*J* _max_	*K* _max_	Int. Sum (cm/Molecule)		
					RV ^d^	*E_emp_* ^e^					
									Exp.	Var.	Exp./Var.
*ν*_2_ + 2*ν*_5_ + *ν*_6_	*E*	6236.3581 (0.54) 6277.7417 (0.79)	1298	5.82 × 10^−22^	532	439	11	9	5.11 × 10^−22^	5.53 × 10^−22^	0.92
*ν*_2_ + 2*ν*_5_ + *ν*_6_	*A* _1_	6254.1267 (0.84)	348	3.06 × 10^−23^	135	71	9	8	2.19 × 10^−23^	2.27 × 10^−23^	0.96
*ν*_1_ + *ν*_2_ + *ν*_6_	*E*	6298.5429 (0.61)	388	4.58 × 10^−22^	189	173	11	10	4.06 × 10^−22^	4.49 × 10^−22^	0.90
*ν*_2_ + *ν*_4_ + *ν*_6_	*A* _1_	6336.9540 (−0.76)	433	4.83 × 10^−22^	246	216	11	10	5.59 × 10^−22^	4.69 × 10^−22^	1.19
*ν*_2_ + *ν*_4_ + *ν*_6_	*A* _2_	6356.4186 (0.67)	321	1.03 × 10^−22^	146	128	11	9	1.19 × 10^−22^	9.61 × 10^−23^	1.24
*ν*_2_ + *ν*_4_ + *ν*_6_	*E*	6347.7418 (0.75)	758	4.21 × 10^−22^	386	310	11	9	4.18 × 10^−22^	3.96 × 10^−22^	1.06
3*ν*_2_	*A* _1_	6426.9446 (0.53)	289	2.07 × 10^−21^	212	212	12	11	1.95 × 10^−21^	2.05 × 10^−21^	0.95
*ν*_1_ + 3*ν*_6_	*A* _1_	6431.4574 (−0.62)	136	1.00 × 10^−22^	61	30	9	5	1.89 × 10^−22^	9.35 × 10^−23^	2.02
*ν*_1_ + 3*ν*_6_	*E*	6417.2823 (0.59)	268	2.84 × 10^−23^	110	73	9	9	2.65 × 10^−23^	2.48 × 10^−23^	1.07
2ν_5_ + 3*ν*_6_	*A* _1_	6376.2454 (0.50) 6374.0419 (−0.69)	226	1.54 × 10^−23^	20	15	12	5	1.02 × 10^−23^	9.70 × 10^−24^	1.05
2*ν*_5_ + 3*ν*_6_	*A* _2_	6378.2876 (0.71) 6375.1935 (−0.52)	364	2.30 × 10^−23^	38	21	8	6	2.30 × 10^−23^	1.57 × 10^−23^	1.46
2*ν*_5_ + 3*ν*_6_	*E*	6431.1383 (−0.63)	1351	7.05 × 10^−23^	50	33	11	7	2.80 × 10^−23^	3.01 × 10^−23^	0.93
		6396.3973 (0.82)									
		6368.9869 (−0.62)									
*ν*_4_ + 3*ν*_6_	*A* _1_	6460.5624 (0.74)	269	1.35 × 10^−23^	24	24	11	10	4.63 × 10^−24^	4.12 × 10^−24^	1.12
*ν*_4_ + 3*ν*_6_	*A* _2_	6478.6942 (−0.83)	212	6.43 × 10^−24^	4	4	9	7	1.10 × 10^−24^	1.10 × 10^−24^	1.00
*ν*_4_ + 3*ν*_6_	*E*	6464.6815 (0.66)	1092	1.33 × 10^−22^	199	162	12	5	7.55 × 10^−23^	7.49 × 10^−23^	1.01
		6480.2012 (0.74)									
		6482.9860 (0.67)									
*ν*_3_ + *ν*_5_ + 3*ν*_6_	*E*	6255.4872 (0.60)	1009	9.81 × 10^−23^	177	105	10	6	6.13 × 10^−23^	6.72 × 10^−23^	0.91
		6219.9269 (−0.83)									
		6224.6451 (−0.59)									
2*ν*_3_ + *ν*_5_ + 2*ν*_6_	*A* _2_	6353.4585 (−0.62)	252	3.30 × 10^−23^	43	33	7	5	2.70 × 10^−23^	1.85 × 10^−23^	1.46
*ν*_2_ + *ν*_3_ + 2*ν*_5_	*E*	6427.0159 (−0.87)	404	8.62 × 10^−23^	28	22	11	3	7.82 × 10^−23^	4.75 × 10^−23^	1.65
*ν*_2_ + *ν*_3_ + *ν*_4_	*E*	6505.0296 (−0.81)	378	8.89 × 10^−23^	108	82	9	7	4.74 × 10^−23^	4.71 × 10^−23^	1.01
*ν*_2_ + 2*ν*_3_ + *ν*_5_	*E*	6247.0639 (0.75)	686	8.84 × 10^−23^	216	154	11	7	6.44 × 10^−23^	7.02 × 10^−23^	0.92
		**Total**	**10,482**	**4.93 × 10^−21^**	**2924**	**2307**			**4.62 × 10^−21^**	**4.54 × 10^−21^**	**1.02**

Notes: ^a^ Variational value of the band center with, within parentheses, the coefficient with maximum amplitude in the expansion of the eigenstate in the normal mode basis at *J* = 0. In some cases, the vibrational labeling is ambiguous and corresponds to several distinct upper vibrational states. ^b^ Total number of transitions of the considered band in the variational line list. ^c^ The intensity cut-off of the variational list at 296 K is 10^−26^ cm/molecule. ^d^ Rovibrational assignment obtained by comparison with the variational list. ^e^ Empirical value of the lower-state energy obtained by the 2*T*-method.

**Table 2 molecules-29-05276-t002:** Statistics of the transitions assigned to twenty bands in the absorption spectrum of ^12^CH_3_D between 6090 and 6530 cm^−1^ at 294 K and comparison of the measured and calculated intensities.

		Variational			Assigned Lines						
Band	Sym.	Center (cm^−1^) ^a^	Nb. ^b^	Int. Sum 296 K ^c^ (cm/Molecule)	Number		*J* _max_	*K* _max_	Int. Sum (cm/Molecule) 294 K		
					RV ^d^	*E_emp_* ^e^			Exp.	Var.	Exp./Var.
*ν*_2_ + 2*ν*_5_ + *ν*_6_	*E*	6236.3581 (0.54)	2315	5.64 × 10^−22^	537	449	16	11	4.22 × 10^−22^	4.19 × 10^−22^	1.01
		6277.7417 (0.79)									
*ν*_2_ + 2*ν*_5_ + *ν*_6_	*A* _1_	6254.1267 (0.84)	523	4.21 × 10^−23^	80	72	10	8	2.51 × 10^−23^	1.91 × 10^−23^	1.31
*ν*_1_ + *ν*_2_ + *ν*_6_	*E*	6298.5429 (0.61)	915	3.71 × 10^−22^	231	183	14	12	2.89 × 10^−22^	2.87 × 10^−22^	1.01
*v*_2_ + *ν*_4_ + *ν*_6_	*A* _1_	6336.9540 (−0.76)	818	3.71 × 10^−22^	301	229	14	12	4.25 × 10^−22^	3.30 × 10^−22^	1.29
*ν*_2_ + *ν*_4_ + *ν*_6_	*A* _2_	6356.4186 (0.67)	700	1.68 × 10^−22^	163	137	12	10	1.63 × 10^−22^	1.20 × 10^−22^	1.36
*ν*_2_ + *ν*_4_ + *ν*_6_	*E*	6347.7418 (0.75)	1382	3.83 × 10^−22^	375	313	12	10	3.25 × 10^−22^	2.81 × 10^−22^	1.16
3*ν*_2_	*A* _1_	6426.9446 (0.53)	508	1.76 × 10^−21^	282	221	14	12	1.65 × 10^−21^	1.69 × 10^−21^	0.98
*ν*_1_ + 3*ν*_6_	*A* _1_	6431.4574 (−0.62)	167	5.34 × 10^−23^	30	30	9	5	8.46 × 10^−23^	4.79 × 10^−23^	1.77
*ν*_1_ + 3*ν*_6_	*E*	6417.2823 (0.59)	490	4.21 × 10^−23^	87	75	15	9	2.62 × 10^−23^	1.78 × 10^−23^	1.47
2ν_5_ + 3*ν*_6_	*A* _1_	6376.2454 (0.50) 6374.0419 (−0.69)	237	2.95 × 10^−23^	21	13	15	6	2.14 × 10^−23^	1.85 × 10^−23^	1.16
2*ν*_5_ + 3*ν*_6_	*A* _2_	6378.2876 (0.71) 6375.1935 (−0.52)	340	2.33 × 10^−23^	22	20	10	6	7.01 × 10^−24^	4.92 × 10^−24^	1.42
2*ν*_5_ + 3*ν*_6_	*E*	6431.1383 (−0.63)	1337	7.83 × 10^−23^	35	31	11	7	1.11 × 10^−23^	1.13 × 10^−23^	0.98
		6396.3973 (0.82)									
		6368.9869 (−0.62)									
*ν*_4_ + 3*ν*_6_	*A* _1_	6460.5624 (0.74)	452	1.12 × 10^−22^	51	29	13	11	8.13 × 10^−23^	7.81 × 10^−23^	1.03
*ν*_4_ + 3*ν*_6_	*A* _2_	6478.6942 (−0.83)	265	1.87 × 10^−23^	5	4	9	7	9.85 × 10^−24^	1.00 × 10^−23^	0.99
*ν*_4_ + 3*ν*_6_	*E*	6464.6815 (0.66)	1495	1.46 × 10^−22^	216	171	14	6	7.53 × 10^−23^	6.32 × 10^−23^	1.19
		6480.2012 (0.74)									
		6482.9860 (0.67)									
*ν*_3_ + *ν*_5_ + 3*ν*_6_	*E*	6255.4872 (0.60)	911	8.33 × 10^−23^	107	103	10	6	3.52 × 10^−23^	2.80 × 10^−23^	1.26
		6219.9269 (−0.83)									
		6224.6451 (−0.59)									
2*ν*_3_ + *ν*_5_ + 2*ν*_6_	*A* _2_	6353.4585 (−0.62)	258	2.32 × 10^−23^	36	33	7	5	1.67 × 10^−23^	1.07 × 10^−23^	1.56
*ν*_2_ + *ν*_3_ + 2*ν*_5_	*E*	6427.0159 (−0.87)	541	7.16 × 10^−23^	20	19	13	9	3.99 × 10^−23^	2.63 × 10^−23^	1.52
*ν*_2_ + *ν*_3_ + *ν*_4_	*E*	6505.0296 (−0.81)	573	6.11 × 10^−23^	106	87	11	7	2.84 × 10^−23^	2.57 × 10^−23^	1.11
*ν*_2_ + 2*ν*_3_ + *ν*_5_	*E*	6247.0639 (0.75)	1116	1.23 × 10^−22^	184	153	12	8	6.89 × 10^−23^	6.22 × 10^−23^	1.11
		**Total**	**15,345**	**4.53 × 10^−21^**	**2889**	**2372**			**3.80 × 10^−21^**	**3.55 × 10^−21^**	**1.07**

Notes: ^a^ Variational value of the band center with, within parentheses, the coefficient with maximum amplitude in the expansion of the eigenstate in the normal mode basis at *J* = 0. In some cases, the vibrational labeling is ambiguous and corresponds to several distinct upper vibrational states. ^b^ Total number of transitions of the considered band in the variational line list. ^c^ The intensity cut-off of the variational list at 296 K is 10^−26^ cm/molecule. ^d^ Rovibrational assignment obtained by comparison with the variational list. ^e^ Empirical value of the lower-state energy obtained by the 2*T*-method.

**Table 3 molecules-29-05276-t003:** Statistics of the transitions assigned to twenty bands in the absorption spectrum of ^12^CH_3_D between 6190 and 6530 cm^−1^ at 294 K and 81 K and comparison of the measured lines with assigned transitions and transitions with *E_emp_*.

	294 K			81 K		
	Nb.	Int. Sum (cm/Molecule)	Proportion (%)	Nb.	Int. Sum (cm/Molecule)	Proportion (%)
All	9344	4.90 × 10^−21^	100	8135	5.12 × 10−21	100
*E_emp_*	4395	4.12 × 10^−21^	84.08	4395	4.84 × 10−21	94.53
RV assign. ^a^	2779	3.77 × 10^−21^	76.94	2809	4.61 × 10−21	90.04
RV assign. + *E_emp_*	2292	3.56 × 10^−21^	72.65	2230	4.48 × 10−21	87.50
Only *E_emp_*	2099	5.60 × 10^−22^	11.43	2164	3.58 × 10−22	6.99
Only RV assign.	487	2.12 × 10^−22^	4.33	581	1.35 × 10−22	2.64
Nothing	4466	5.74 × 10^−22^	11.71	3160	1.49 × 10−22	2.91

Notes: All the statistics in this table correspond to the spectral region between 6190 and 6530 cm^−1^ for the two temperatures. ^a^ Rovibrational assignment based on the TheoReTS variational list.

## Data Availability

The data are provided as Appendix A.

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
