# Peer review of "The CH_3_D Absorption Spectrum Near 1.58 μm: Extended Line Lists and Rovibrational Assignments"

_molecules, 2024, doi:10.3390/molecules29225276_

Round 1
Reviewer 1 Report
Comments and Suggestions for Authors
The article under the title “The CH3D absorption spectrum near 1.58 μm: extended line lists and rovibrational assignments.” by Fatallah presents results on the assignment of rotovibrational bands of CH3D. The authors presented details on the experimental procedure and selection of bands, with accompanying theoretical calculations. The article could be of potential interest to the readers of the Molecules, although there are some points that should be addressed before the final decision. Therefore, my recommendation is MAJOR REVISION.
The authors should answer the following:
1. The authors should verify the use of the Voigit function for the line shape by citing relevant literature
2. Figure 9 should be more clear
3. The authors should discuss if differences in values are more pronounced for higher or lower wavenumber frequency region
4. What is the chemical rationale of the empirical correction of the rotational energies in the region
5. The details on calculations are missing and the verification of their applicability on similar systems
Reviewer 2 Report
Comments and Suggestions for Authors
The manuscript provides a comprehensive analysis of the CH3D (monodeuterated methane) absorption spectrum in the 1.58 μm region, which is crucial for studying the atmospheres of giant planets and Titan. The authors have collected extensive spectral data using high-sensitivity differential absorption spectroscopy (DAS), including over 11,000 lines. The integration of experimental data with theoretical predictions from the TheoReTS variational line list is commendable, and the systematic validation of rovibrational assignments using Ground State Combination Difference (GSCD) relations further strengthens the accuracy of the results. Additionally, the identification of 15 new spectral bands adds significant value, offering a well-rounded and in-depth exploration of CH3D's absorption characteristics for planetary applications. The study makes a valuable contribution to the field, though there are still areas for improvement.
1. While the agreement between experimental and theoretical data is generally good, there are notable discrepancies, especially for the 3v2 band where the vibrational energy calculated by TheoReTS is underestimated by up to 2 cm^-1. The authors should provide more in-depth discussion regarding the possible reasons behind these deviations, such as insufficient convergence in high J values or local resonance interactions.
2. The paper briefly mentions the interaction between the 3v2 and v1+3v6 A1 bands. A more detailed analysis of how this interaction affects energy shifts and intensity variations would enhance the paper. Additionally, the practical implications of this interaction for planetary atmosphere modeling could be discussed further.
3. Although the authors assigned most of the strong and medium-intensity lines, some lines remain unassigned. Providing hypotheses on the potential origin of these unassigned lines, such as unidentified isotopologues or local perturbations, would be helpful for future studies.
This paper makes a valuable contribution to the study of CH3D absorption spectra, especially in planetary atmosphere research. The authors successfully combine experimental and theoretical approaches, providing a detailed analysis with extensive data. Addressing the areas for improvement mentioned above would further enhance the clarity and impact of the paper.
Round 2
Reviewer 1 Report
Comments and Suggestions for Authors
The authors have answered all of the questions properly, the manuscript is acceptable for publication in the present state.